# OpenReview forum: "AutoRedTeamer: Autonomous Red Teaming with Lifelong Attack Integration"
_NeurIPS.cc/2025/Conference — NeurIPS 2025 poster_

### Official Review · Reviewer_XEiq · 2025-06-24

**Clarity:** 1
**Significance:** 2
**Originality:** 2
**Rating:** 2
**Confidence:** 4

**Summary:**

This paper proposes a black-box, automated red teaming framework that integrates a multi-agent architecture with a memory-guided attack selection mechanism to support continuous discovery and adaptation of new attack vectors. The framework features two key agents: a red teaming agent capable of generating and executing test cases based solely on high-level risk categories, and a strategy proposer agent that autonomously identifies and incorporates novel attack strategies by analyzing recent research developments.

**Questions:**

See above in weakness.

**Ethical Concerns:**

["Major Concern: Safety and security"]

**Final Justification:**

While I appreciate the rebuttal from the authors, I still find the key process of literature mining, which is acknowledged by the authors as the main contribution, unclear and insufficiently discussed or evaluated in the current version. I remain uncertain about how attack vectors are mined from the literature. In the first response, the authors state that embedding similarity is used to compare the literature with current attacks, but it is unclear how the embeddings of long paper contexts are processed and how key attack vectors are identified in such a long context. They also do not explain how performance metrics are incorporated into the scoring process.

Furthermore, in subsequent comments, the authors revise their description of the process, stating that an LLM is used to evaluate only the abstract of the paper to extract attacks. This raises additional concerns, as abstracts typically provide high-level overviews and rarely contain specific attack vectors or detailed descriptions, which makes me doubt the effectiveness of this mining approach and the uniqueness of constructed attacks.

Overall, I believe there remains a substantial gap in the mining process as presented in the current version, and it will require another round of revision. Therefore, I maintain my score.

**Limitations:**

See above in weakness.

**Quality:**

2

**Strengths And Weaknesses:**

**Strength**

This paper proposes a simple and effective red-teaming framework that operates in a black-box setting, making it applicable to both open-source and closed-source large language models.

**Weakness**

1. Unclear attacker proposer Design in Sec. 3.2. The design of the Attacker Proposer is not clearly described. Specifically, there is insufficient detail on how papers are transformed into executable attack code. Since this component is central to the attack capabilities of the proposed framework, more information is needed regarding how the implementation ensures quality and effectiveness. For example, what specific criteria are used to evaluate the novelty of an attack?
2. It is unclear what types of attacks the agent actually implements. The main body seems to indicate that the agent can generate executable plug-and-play attack code (e.g., GCG [1]) or scripts that can be run automatically. However, based on the examples in Section E, it appears that the system primarily stores natural language descriptions of harmful prompt types, rather than actual attack code. These are closer to harmful prompt templates than to specific novel attacks.
3. The authors state that “the agent also generates additional attack proposals by identifying core principles from the analyzed papers and combining them in new ways.” However, there is no detailed explanation of how such combinations are implemented.
4. The description in Sec. 3.5 “by tracking each test case’s effectiveness” is unclear. It is unclear whether this effectiveness is evaluated by the attacker agent itself, or by observing the response from the target model. More clarity is needed on how this evaluation is carried out.
5. The cost and purpose of validation set filtering. The framework includes a validation phase before actual red-teaming, which raises questions about cost and purpose. Since red-teaming aims to uncover vulnerabilities in a target model, using a validation step to pre-filter attack cases may add unnecessary computational cost and may also bias the resulting test cases. This pre-filtering may discard attacks that might be effective on certain target models.
6. The novelty is limited. The proposed “Attack Memory” concept resembles prior work on skill libraries (e.g., [2], [3]), and the multi-agent red-teaming approach is similar to frameworks explored in [3], [4]. As such, the novelty of the contributions appears limited.
7. The evaluation is restricted to only 4 LLMs. A broader evaluation over more models would strengthen the empirical support for the framework’s generality and effectiveness.
8. A small point is that the paper aims to propose an automatic red-teaming framework; it is unclear why it still requires user-provided seed attacks. Ideally, the framework should also automate this seed generation step or incorporate it into the overall system design.

[1] Zou, Andy, et al. "Universal and transferable adversarial attacks on aligned language models." arXiv preprint arXiv:2307.15043 (2023).

[2] Fan, Linxi, et al. "Minedojo: Building open-ended embodied agents with internet-scale knowledge." Advances in Neural Information Processing Systems 35 (2022): 18343-18362.

[3] Liu, Xiaogeng, et al. "Autodan-turbo: A lifelong agent for strategy self-exploration to jailbreak llms." arXiv preprint arXiv:2410.05295 (2024).

[4] BENCHMARK, JUDGE. "JAILJUDGE: AComprehensive JAILBREAK JUDGE BENCHMARK WITH MULTI-AGENT ENHANCED EXPLANATION EVALUATION FRAMEWORK."

---

> ### Author Rebuttal · Authors · 2025-07-31
>
> We thank you for your detailed technical analysis and address each concern systematically to clarify our framework's capabilities and contributions.
>
> *Q1 - Attacker Proposer Design Clarity (Sec. 3.2)*
>
> Our Attack Proposer transforms papers into executable code through a structured pipeline: (1) Paper Retrieval: QUERYPAPERS uses Semantic Scholar API with targeted queries like "jailbreaking language models 2024", (2) Scoring Mechanism: Papers receive composite scores based on novelty (semantic similarity <0.7 to existing attacks using sentence embeddings), feasibility (black-box implementability), and reported performance metrics, (3) Attack Generation: GENERATEPROPOSALS extracts implementable techniques and combines core principles from multiple papers, (4) Code Implementation: IMPLEMENTATTACK converts proposals to Python classes inheriting our base interface, with complete validation on HarmBench subsets requiring >30% ASR before integration.
>
> *Q2 - Attack Implementation Types*
>
> Your observation reveals an important clarification about red teaming attacks. The Cultural References example in Section E is indeed an adaptive executable attack implemented in natural language - similar to how PAIR adaptively generates prompts through code but produces natural language outputs. This is the standard approach: most successful red teaming attacks operate in natural language (PAIR, TAP, our Cultural References, Technical Slang), while only a few are pure templates (like Pliny). Our framework implements the full spectrum: (1) Adaptive natural language attacks: Cultural References, TechnicalSlang, RolePlay - these use algorithmic logic to generate contextually appropriate natural language prompts, (2) Algorithmic transformations: ArtPrompt (ASCII encoding), encoding-based attacks, (3) Template-based attacks: Pliny, HumanJailbreaks. The key innovation is that our Attack Proposer can automatically implement new adaptive attacks from research literature, not just store static templates.
>
> *Q3 - Attack Combination Implementation*
>
> Combinations are implemented through sequential transformation pipelines. The Strategy Designer selects attack sequences applied sequentially (e.g., TechnicalSlang→Pliny: first transforms "create malware" to "develop persistent threat vectors", then applies Pliny jailbreak wrapper). Our memory system tracks transition probabilities between attacks (Figure 11), learning effective sequences rather than random combinations.
>
> *Q4 - Effectiveness Evaluation Process*
>
> This evaluation is performed by observing target model responses, not self-assessment by attackers. The process: (1) Attack transforms test case p→p', (2) Target model generates response r=LLM(p'), (3) Our EVALUATOR scores r on 1-10 harmfulness scale, (4) Success (score >threshold) updates memory with combination effectiveness. This is standard red teaming evaluation methodology.
>
> *Q5 - Validation Set Filtering*
>
> The validation step is both computationally cheap and critical in our setup. At 80 queries, it's negligible compared to baselines (60,000 for AutoDAN-Turbo), but essential because most proposed attacks fail. Without validation, these attacks would pollute our library and waste evaluation budget on ineffective strategies. Our 30% ASR threshold ensures only genuinely effective attacks enter the system, preventing resource waste during actual red teaming. This quality control is a key design principle that differentiates our approach from naive attack collection. A stronger way to validate attacks besides a thershold could be an interesting avenue to explore.
>
> *Q6 Addressing Novelty Concerns*
>
> While building on existing concepts, our specific contributions represent genuine innovations: (1) Automated literature mining for attack discovery - automatically extracting and implementing attacks from research papers, (2) Memory-guided attack combination strategy - discovering synergistic attack sequences through transition matrix learning, (3) End-to-end risk category evaluation - operating directly from high-level categories to generate diverse test cases, (4) Lifelong attack integration - continuous adaptation through automated research analysis. These represent a paradigm shift from static to adaptive red teaming.
>
> *Q7-8 Evaluation Scope and Seed Requirements*
>
> We acknowledge the 4-LLM limitation reflects resource constraints. Regarding seed requirements: seeds are useful but not necessary. Our framework operates fully autonomously from risk categories (demonstrated in AIR evaluation), and Table 4 shows our system achieves 0.78 ASR using only automatically discovered attacks (without any human-initialized attacks).
> We appreciate your thorough analysis and believe these clarifications address the core concerns while highlighting our framework's genuine innovations.

---

> > ### Comment · Reviewer_XEiq · 2025-08-05
> >
> > Thanks for the detailed reply. While some of my concerns have been addressed, others remain unresolved.
> >
> > First, I am still unclear about the scoring mechanism. The current description remains somewhat high-level, and its effectiveness is not well justified. There appear to be no concrete examples or detailed evaluations of how the paper scoring step works. I understand that this step may be performed using a magic LLM, but I am particularly interested in how effective this is in practice. For instance, the authors mention using sentence embeddings for semantic similarity, but how are the key method sentences identified for a given paper? Additionally, the rebuttal mentions incorporating performance metrics into the scoring process, but there is no concrete explanation of how these metrics are used to determine the score. Since code is not provided, a more detailed description would greatly help clarify this process.
> >
> > Second, I remain doubtful about the novelty of the work. The main contributions listed by the authors appear to have been explored in prior literature. What I considered to be the most unique contribution, automatic paper mining, is still not clearly explained. Both the paper and the rebuttal lack sufficient detail and evaluation about this step, which makes it difficult to assess its novelty or technical depth.

---

> > > ### Author Response · Authors · 2025-08-05
> > >
> > > Thank you for the continued engagement and specific feedback. We address your remaining concerns with additional technical detail:
> > >
> > > *Scoring Mechanism Clarification*
> > >
> > > You're correct to seek more concrete detail. I apologize for the inconsistency in the previous response - the scoring process is primarily LLM-based rather than embedding-based. Specifically: (1) Method Identification: Our LLM analyzes paper abstracts to identify core methodological contributions using prompts like "Extract the main attack technique described in this abstract," (2) Novelty Assessment: The LLM compares extracted methods against our existing attack library descriptions and provides novelty scores on a 0-10 scale with justification, (3) Performance Integration: We extract reported ASR numbers from abstracts using both regex patterns and LLM parsing for cases where metrics are described in natural language, (4) Composite Score: Final score combines LLM-assessed novelty, extracted performance metrics, and LLM-evaluated feasibility for black-box implementation. For example, when analyzing a paper on "cultural context manipulation," the LLM would assess its distinctiveness from existing attacks in our library and provide a novelty justification. The crucial verification step is the actual evaluation of the code implementation of the attack afterwards. We will add a more detailed description of this in the updated paper.
> > >
> > > *Novelty Clarifications*
> > >
> > > Thank you for acknowledging the potential novelty of our automated paper mining contribution. We appreciate this recognition and would like to clarify how AutoRedTeamer fundamentally differs from the cited works in both approach and technical contributions.
> > >
> > > - AutoDAN-Turbo [3] vs. AutoRedTeamer: While AutoDAN-Turbo discovers attacks through evolutionary self-exploration of prompt variations, AutoRedTeamer introduces a fundamentally different paradigm: automated extraction and implementation of attacks from research literature. AutoDAN-Turbo evolves prompts through mutation and selection; we automatically parse academic papers, extract attack methodologies, and generate executable code implementations. This enables our system to immediately incorporate cutting-edge attacks as they appear in research, rather than hoping to rediscover them through evolution.
> > > - Skill Libraries [2] vs. Attack Memory: Generic skill libraries store independent skills for reuse. Our memory system specifically tracks attack combination synergies through transition matrices (Fig. 11), learning which attack sequences amplify effectiveness. For example, we discover that Pliny→ArtPrompt achieves 0.83 ASR while individual attacks achieve only 0.63 and 0.40 respectively. This transition-based learning for discovering attack synergies is novel in the red teaming context.
> > > - JailJudge [4] vs. AutoRedTeamer: JailJudge focuses on evaluating jailbreak attempts with multi-agent judgment, while we focus on generating attacks through automated research analysis. These are complementary but fundamentally different problems. In general, we believe multi-agent approaches to red teaming are a broad research area, and it is worth exploring different multi-agent designs.
> > >
> > > *Automated Paper Mining clarification*
> > >
> > > The automated paper mining that you identified as potentially novel involves several non-trivial technical contributions:
> > >
> > > - Semantic parsing of attack methodologies from unstructured academic text into structured attack specifications and new attack proposals
> > > - Automated code synthesis that transforms paper descriptions into executable attack classes with our standardized interface
> > > - Black-box adaptation ensuring paper attacks work without model internals access
> > > - Quality validation through automated testing before integration
> > >
> > > As demonstrated in Section C.2 and Appendix E, our system successfully implemented 8 novel attacks from recent papers, including our "Cultural References" attack that achieved 0.48 ASR. This capability to continuously expand attack capabilities through automated research analysis represents a paradigm shift from static attack libraries or evolutionary approaches.
> > >
> > > We will enhance the paper to better emphasize these unique contributions and provide more technical detail on the paper mining pipeline that enables this novel capability.

---

### Official Review · Reviewer_GG1W · 2025-07-02

**Clarity:** 3
**Significance:** 3
**Originality:** 4
**Rating:** 5
**Confidence:** 3

**Summary:**

Large Language Models (LLMs) are continually evolving, and each new iteration may introduce vulnerabilities to both existing and emerging threats. Traditionally, identifying these vulnerabilities has relied on manual effort or human-in-the-loop approaches. This paper presents a novel autonomous red-teaming multi-agent agent framework that can be continuously updated with new agents as threats evolve. The framework enables continuous testing to proactively uncover vulnerabilities in LLMs. Additionally, it employs a memory-guided strategy, leveraging prior threat signatures and their ordering to intelligently launch targeted vulnerability detection agents.

**Questions:**

1. Are there theoretical limitations or deeper ML questions that may not have been fully addressed due to a stronger focus on empirical performance?

2. Include a dedicated “Limitations” section instead of embedding them in the conclusion and appendix, to improve clarity and completeness.

3. Can you provide a playbook for integrating new agents into this red teamer framework by those outside of the authors/researchers. This will improve the usefulness of the framework and its impact.

**Ethical Concerns:**

["NO or VERY MINOR ethics concerns only"]

**Final Justification:**

I am in strong favor of supporting this paper. My initial instincts were reaffirmed by the discussions during the rebuttal. I strongly support accepting the paper.

**Limitations:**

Yes.  Include a dedicated “Limitations” section instead of embedding them in the conclusion and appendix, to improve clarity and completeness.

**Paper Formatting Concerns:**

There are no major paper formatting concerns. A nit-- In the appendix choice of the background to highlight the foreground text  may need some experimentation.

**Quality:**

3

**Strengths And Weaknesses:**

Strengths: The paper presents a major advance over the state-of-the-art by enabling autonomous and continuous vulnerability detection in LLMs without requiring a human-in-the-loop.

A key innovation is the memory-guided strategy for ordering and launching vulnerability detectors, which outperforms random and heuristic baselines. This is demonstrated clearly using robust metrics such as attack success rate (ASR), number of queries per successful attack, and execution time.

The use of Harmbench—a well-recognized benchmark of adversarial prompts—is a plus. Dedicated agents are used  for each vulnerability class.

The agent pipeline (identify → assess likelihood → analyze risk → launch attack →evaluate attack success) is intuitive, well-structured and scalable as new attacks emerge.

The framework is validated on high-profile LLMs including LLAMA3.0-70B, ChatGPT-4o, Mixtral-8x7B, and Claude 3.5 Sonnet. Notably, it achieves orders-of-magnitude faster query times compared to prior methods, while maintaining comparable or superior success rates.

The appendix provides extensive ablation studies, illustrating the impact of the proposed memory-guided discovery approach across a broad set of harm categories (about a dozen).

Overall, the paper is comprehensive, well-executed, and offers meaningful contributions and novelty.

Weaknesses: There are few apparent weaknesses. However, due to a limited background in ML theory, there may be deeper theoretical aspects that are not fully assessed here. While the current limitations are mentioned in the conclusion and appendix, they should be clearly presented in a standalone “Limitations” subsection to improve clarity and transparency.

---

> ### Author Rebuttal · Authors · 2025-07-31
>
> We thank you for recognizing our work as "a major advance over the state-of-the-art" with "meaningful contributions and novelty." We address your suggestions below:
>
> *Q1: Theoretical limitations and deeper ML considerations*
>
> You correctly identify theoretical gaps in our approach: (1) our memory-guided selection lacks formal convergence guarantees, operating heuristically rather than through principled multi-armed bandit approaches, (2) we cannot provide theoretical bounds for attack transferability across models, and (3) our LLM-based evaluation inherits limitations of using LLMs as judges, including potential systematic biases. While our focus is empirical effectiveness, we acknowledge these theoretical foundations would strengthen the work.
>
> *Q2: Dedicated Limitations section*
>
> Absolutely - we will add a standalone "Limitations" section as you suggest. Currently these are scattered across conclusion and appendix, reducing clarity. Key limitations include dependence on LLM quality for attack implementation, no formal attack space coverage guarantees, computational costs for comprehensive search, and potential biases in literature analysis.
>
> *Q3: Integration playbook for new agents*
>
> Excellent suggestion for broader impact. Our modular design enables straightforward extension: (1) new attack agents inherit from our base interface (Section 3.3) with standardized transformation logic, (2) new evaluation agents follow our JSON format (Appendix G), and (3) new discovery agents can extend the Strategy Proposer with additional research sources beyond Semantic Scholar. We will include a detailed integration guide with code templates in the camera-ready version.
>
> We commit to adding the dedicated Limitations section and integration playbook to strengthen the paper's completeness and facilitate community contributions

---

> > ### Comment · Reviewer_GG1W · 2025-08-05
> >
> > Thank you for the responses. I am satisfied with them and I don't have any more questions.

---

> > > ### Comment · Reviewer_GG1W · 2025-08-09
> > >
> > > I followed the discussions including author  responses, and these reinforced my assessment of contributions that this work makes. I am happy with the initial score and will maintain this score!

---

### Official Review · Reviewer_TAJ9 · 2025-07-02

**Clarity:** 3
**Significance:** 3
**Originality:** 3
**Rating:** 5
**Confidence:** 4

**Summary:**

This paper presents AutoRedTeamer, an autonomous framework designed to evaluate the robustness of LLMs against jailbreak attacks. The system continuously learns new attack strategies by analyzing research papers, implementing and evaluating their effectiveness, and maintaining a memory system that tracks the success rates of various attack techniques. The framework comprises two main components: a strategy proposer that builds an evolving attack toolbox from academic literature, and a red teaming agent that uses this toolbox to evaluate the safety of target LLMs. Experiments on the HarmBench dataset, conducted across four different LLMs and compared against four optimization-based and three static template-based baseline attacks, demonstrate that AutoRedTeamer outperforms existing approaches. An ablation study further highlights the importance of the memory component to the overall system's effectiveness.

**Questions:**

- Are the successful attacks dominated by a particular subset of techniques discovered through the literature review?
- If so, are these dominant techniques consistent across different LLMs or different categories of harmful content (as shown on the x-axis in Figure 3)?
- Can you also share any insights into why AutoRedTeamer's is particularly successful on Claude-3.5-Sonnet?

**Ethical Concerns:**

["NO or VERY MINOR ethics concerns only"]

**Final Justification:**

The authors have adequately addressed my comments and I will maintain my already favorable score.

**Limitations:**

Limitations are adequately discussed.

**Paper Formatting Concerns:**

No formatting concerns.

**Quality:**

4

**Strengths And Weaknesses:**

I enjoyed reading this paper and believe that it presents a promising autonomous end-to-end system for evaluating the robustness of LLMs against jailbreak attacks. While the paper does not introduce novel attack techniques of its own, it makes a valuable contribution by integrating a wide range of known attacks and evaluating weaknesses of LLMs to specific attacks. The evaluation shows the effectiveness of the proposed techniques and I also appreciate the inclusion of the ablation studies that shed light on the importance of the proposed memory system.

I would have liked to see more analysis and discussion of the evaluation results. For instance, AutoRedTeamer performs particularly well on Claude-3.5-Sonnet, are there specific attack strategies that drive this success? Understanding these patterns could be instrumental in developing defenses. I encourage the authors to add further discussion on this point to the paper.

---

> ### Author Rebuttal · Authors · 2025-07-31
>
> We thank you for your positive assessment, noting that our work presents "a promising autonomous end-to-end system" with "valuable contribution" and "effective evaluation." We appreciate your recognition of our ablation studies and address your insightful questions below:
>
> *Q1: Attack strategy patterns driving Claude-3.5-Sonnet success*
> Our analysis reveals that AutoRedTeamer's exceptional performance on Claude-3.5-Sonnet (0.28 ASR vs. near-zero for baselines) stems from its adaptive attack combination strategy. As shown in Figure 8, the most effective combinations involve synergistic attacks: Pliny+ArtPrompt (0.83 ASR), TechnicalSlang+Pliny, and combinations leveraging both encoding-based and social engineering approaches. Claude-3.5-Sonnet appears particularly vulnerable to multi-vector attacks that combine authoritative framing (like our Cultural References attack from Section E) with technical encoding, whereas single-vector approaches (used by baselines) fail to bypass its robust safety filters.
>
> *Q2: Consistency of dominant techniques across models and content categories*
> Figure 9's transition matrix analysis shows that attack effectiveness varies significantly across models. While combinations involving Pliny and HumanJailbreaks are consistently effective across models, encoding-based attacks (ArtPrompt, TechnicalSlang) show model-specific effectiveness patterns. For content categories, Figure 3 demonstrates that AutoRedTeamer maintains higher ASR across all 43 AIR level-3 categories compared to static approaches, suggesting our memory-guided selection adapts attack strategies to specific content types rather than relying on universally dominant techniques.
>
> *Defense Development Implications:*
> Your point about defense development is valuable. Our findings suggest that effective defenses for Claude-3.5-Sonnet should focus on adaptivity to combinations of attacks and more diverse input perturbations.
>
> We will expand the discussion section in our camera-ready version to include these insights and their implications for both red teaming and defense development, as you suggest.

---

### Official Review · Reviewer_Sd6E · 2025-07-03

**Clarity:** 3
**Significance:** 3
**Originality:** 3
**Rating:** 5
**Confidence:** 4

**Summary:**

This paper proposes a framework based on Large Language Model (LLM) agents for automated red teaming, aiming to assess the security robustness of target LLMs. The framework consists of two main components: (1) Lifelong Attack Integration, which continuously collects and integrates the latest jailbreak strategies from public sources and synthesizes attacks using an attack designer guided by an evaluation mechanism; and (2) Red Teaming Evaluation, which includes a risk analyzer to receive user input, a seed prompt generator, a strategy planner, a memory system to track historical attack attempts, and an automatic evaluation module that scores model responses based on risk and contextual relevance.

The authors evaluate their framework across four LLMs, benchmarking it against existing approaches using attack success rate (ASR) and the number of queries required. Results show higher success rates and improved query efficiency. A comprehensive ablation study further validates the contribution of each component in enhancing red teaming effectiveness.

**Questions:**

Evlaution:

* What is the rationale behind selecting Mixtral and Claude as the primary LLMs in the proposed framework? It is unclear why Claude is only used for attack implementation, while Mixtral appears more centrally involved. A discussion of this design choice, or empirical comparison across different LLMs, would help clarify how various models perform as threat models and their suitability for red teaming.

* Could the authors elaborate on the performance of the attack implementation module? Specifically, how effective is it in generating successful and novel attack prompts, and how does it compare to baselines or other components within the framework?

* It seems like a promising idea to incorporate user input into the proposed framework, particularly for guiding risk assessment or customizing attack objectives. I am curious about the impact of this component: how much does user input contribute to the effectiveness of the red teaming evaluation? Additionally, what would be the expected performance or limitation if the evaluation relied solely on the "attack library" generated by the Attack Strategy Proposer Agent, without incorporating user-driven customization or guidance?

**Ethical Concerns:**

["NO or VERY MINOR ethics concerns only"]

**Limitations:**

yes

**Quality:**

3

**Strengths And Weaknesses:**

Strengths:
The paper is well-written and clearly motivated. It covers and compares with the most relevant studies, and introduces a novel LLM-agent-based framework for red teaming. The evaluation is comprehensive, including both quantitative comparisons and ablation studies that support the effectiveness of the proposed approach.


Weaknesses:
While the paper presents a promising framework, the clarity of the methodology could be improved to enhance overall understanding and reproducibility.

- Figure 1 could benefit from refinement. The naming of agents may cause confusion, as it is not clearly reflected in the figure. Additionally, inconsistencies in font size, visual balance, and component labeling make it difficult to grasp the core elements of the proposed framework at a glance.

- Methodological details are lacking in several key areas:

- - Line 143: It is unclear how the agent assigns a composite score—more explanation or formalization is needed.

- - Line 152: The process by which the agent proposes new attack vectors is not described—what constitutes "new," and how is novelty achieved or evaluated?

- - Line 164: The mechanism for maintaining semantic coherence is underexplained.

- - Line 225: The automatic evaluation procedure lacks sufficient detail. Specifically, how relevance is computed based on model outputs and risk scores.

Greater clarity on these aspects would significantly improve the paper's reproducibility and technical transparency.

---

> ### Author Rebuttal · Authors · 2025-07-31
>
> We thank you for recognizing our framework as "novel" and "well-motivated" with "comprehensive evaluation." We address your specific questions and suggestions below:
>
> *Q1: Rationale for LLM selection (Mixtral vs Claude)*
>
> We use Mixtral-8x22B-Instruct-v0.1 for most modules because it lacks safety training and can be used for attacking other models. Many other LLMs refuse to adopt the roles within the framework.
> Claude-3.5-Sonnet is specifically chosen for attack implementation because: (1) it demonstrates superior code generation capabilities needed for implementing complex attack logic from research papers, and (2) our preliminary experiments showed it produces more reliable attack implementations that pass our validation threshold of 30% ASR. This design choice balances performance with computational efficiency across the framework's different requirements.
>
> *Q2: Attack implementation module performance*
>
> The Attack Strategy Proposer successfully implements 8 new attacks from research literature, as detailed in Section C.2 and demonstrated in our ablation study (Table 4). When using only proposed attacks (without human-developed attacks), AutoRedTeamer achieves 0.78 ASR compared to 0.75 ASR with only human attacks, demonstrating the effectiveness of our automated attack discovery. Section E provides a complete code example of a successful LLM-generated "Cultural References" attack. The validation process ensures quality: attacks must achieve >30% ASR on our validation set before integration.
>
> *Q3: User input impact and attack library vs. user guidance*
>
> This is an interesting suggestion! The framework is flexible to the types of user input. For reproducibility and to match baselines, we tested two types of inputs. The Risk Analyzer (Section 3.4) processes user inputs ranging from high-level categories ("Hate speech") to specific scenarios, as shown in Figure 1. Table 2 demonstrates effectiveness across both modes - the HarmBench evaluation uses specific prompts, while Figure 7 shows results across 314 AIR risk categories using only category names. When relying solely on the attack library without user input, the framework generates test cases directly from risk categories (as in the AIR evaluation), achieving comparable ASR to human-curated benchmarks while requiring no manual test case creation.
>
> *Methodological Clarifications:*
> - Line 143 (composite score): The scoring system uses LLM as a judge to evaluate papers on novelty, distinctiveness from existing attacks, and reported performance. We will make this more explicit.
> - Line 152 (new attack vectors): The agent generates proposals by identifying core principles from analyzed papers and combining them in novel ways, as exemplified by our "Cultural References" attack in Section E. We use the base LLM’s own idea of novelty for this.
> - Line 164 (semantic coherence): Maintained through our validation process requiring 30% ASR. Will update this for clarity
> - Line 225 (automatic evaluation): The relevancy check also uses LLM as a judge, outputting a score from 1 to 10. Will make this more explicit.
>
> We will enhance Figure 1's clarity and provide additional methodological details in the camera-ready version as suggested.

---

> > ### Comment · Reviewer_Sd6E · 2025-08-07
> >
> > Thank you for the detailed and thoughtful responses. I appreciate the clarifications and additional insights, particularly regarding the LLM selection rationale, attack implementation performance, and the role of user input. Your explanations addressed my concerns thoroughly, and I have no further questions at this time.

---

### Decision · Program_Chairs · 2025-09-17

**Decision:**

Accept (poster)

**Comment:**

This paper introduces a novel framework that autonomously discovers and integrates new attack vectors by analyzing recent research and outperforms existing approaches.

The reviewers were divided, but the majority strongly supported the work. Strengths include the novel framework, well-executed and comprehensive evaluation, and its significant advance over the SOTA in enabling continuous and autonomous red teaming. After the rebuttal and discussion, concerns centered on the clarity of the methodology, particularly the paper-mining process.

The authors' rebuttal addressed most of the reviewers' questions, clarifying the technical details and providing more context on how their approach differs from prior work. While one reviewer's specific concerns about the paper-mining methodology were not entirely resolved, the authors promised to adding more detail in the final version.

On balance, the contributions are substantive and timely, thus the AC recommends acceptance, with the suggestion that the camera-ready version provide more explicit details on the prompts and scoring mechanisms used in the paper-mining process to enhance reproducibility.